# Innovation Strategy Selection Facilitates High-Performance Flexible Piezoelectric Sensors

**DOI:** 10.3390/s20102820

**Published:** 2020-05-15

**Authors:** Shengshun Duan, Jun Wu, Jun Xia, Wei Lei

**Affiliations:** Joint International Research Laboratory of Information Display and Visualization, School of Electronic Science and Engineering, Southeast University, Nanjing 210096, China; 220191352@seu.edu.cn (S.D.); xiajun@seu.edu.cn (J.X.); lw@seu.edu.cn (W.L.)

**Keywords:** piezoelectric sensor, material selection, nanocomposite materials, molecular ferroelectric materials, microstructure, chemical doping, piezotronics effect, PVDF

## Abstract

Piezoelectric sensors with high performance and low-to-zero power consumption meet the growing demand in the flexible microelectronic system with small size and low power consumption, which are promising in robotics and prosthetics, wearable devices and electronic skin. In this review, the development process, application scenarios and typical cases are discussed. In addition, several strategies to improve the performance of piezoelectric sensors are summed up: (1) material innovation: from piezoelectric semiconductor materials, inorganic piezoceramic materials, organic piezoelectric polymer, nanocomposite materials, to emerging and promising molecular ferroelectric materials. (2) designing microstructures on the surface of the piezoelectric materials to enlarge the contact area of piezoelectric materials under the applied force. (3) addition of dopants such as chemical elements and graphene in conventional piezoelectric materials. (4) developing piezoelectric transistors based on piezotronic effect. In addition, the principle, advantages, disadvantages and challenges of every strategy are discussed. Apart from that, the prospects and directions of piezoelectric sensors are predicted. In the future, the electronic sensors need to be embedded in the microelectronic systems to play the full part. Therefore, a strategy based on peripheral circuits to improve the performance of piezoelectric sensors is proposed in the final part of this review.

## 1. Introduction

A notable technical trend today is the rapid growth of portable and wearable electronics for applications in personal health care, environmental monitoring and entertainment equipment [1,2]. As the only functional device for data acquisition, sensors have a crucial effect on the performance of the portable and wearable electronics. Therefore, it is necessary to fabricate sensors with wide sensing range and high precision. Moreover, when applied in special environment like human body, sensors are desired to be more flexible and transparent. Thus, substantial flexible resistive and capacitive sensors have been studied and reported, which are widely used in the electronic skin, wearable electronics, personal health care and electronic textiles [3,4,5,6,7,8,9]. However, the demand for extra power supply greatly hinders their applications in flexible microelectronic systems with small size and low consumption. Hence, research on sensors with low-to-zero power consumption and ultrathin structures has great significance.

Piezoelectricity, first discovered in 1880 by Pierre and Jacques Curie [10], can convert mechanical stimulus into electrical signals without extra electric energy. Generally, materials exhibiting the piezoelectric effect (the internal generation of electrical charge resulting from an applied mechanical force) also exhibit the reverse piezoelectric effect (the internal generation of a mechanical strain resulting from an applied electrical field). The coupled equations are written as:(1)$$\{\begin{array}{c} S=s^{E}T+d^{t}E\\ D=dT+\varepsilon^{T}E \end{array}$$
where S is the linearized strain; s is compliance under short-circuit conditions; T is stress; D is the electric flux density (electric displacement); ε is permittivity; E is electric field strength; d is the matrix for the direct piezoelectric effect and $d^{t}$ is the matrix for the converse piezoelectric effect. The superscript E indicates a zero or constant electric field; the superscript T indicates a zero or constant, stress field; and the superscript t stands for transposition of a matrix.

According to whether there are side effects to the environment, piezoelectric materials are classified into lead-containing and lead-free materials [11]. In spite of good piezoelectricity of lead-containing piezoelectric materials, such as ${\mathrm{Pb}}_{x}{\mathrm{Zr}}_{1-x}{\mathrm{TiO}}_{3}$ (PZT), they will gradually be restricted to use in the future, owing to the poisoning effect of Pb on the environment and live things [11,12]. On the other hand, lead-free piezoelectric materials such as zinc oxide (ZnO), barium titanate (${\mathrm{BaTiO}}_{3}$) and polyvinylidene fluoride (PVDF) exhibits many merits, including biocompatibility, good piezoelectric and ferroelectric properties and low cost [13,14,15]. Therefore, the lead-free piezoelectric material is gradually replacing the lead-containing piezoelectric material.

For direct piezoelectric effect, when a mechanical stress is applied to piezoelectric material in a certain direction, polarization occurs inside the piezoelectric material and equivalent opposite charges accumulate at its two faces. The charge accumulating at the surface of the piezoelectric material is given by the piezoelectric effect:(2)$$Q=d_{mn}F_{x}$$
where Q is the charge, $d_{mn}$ is the piezoelectric coefficient, and $F_{x}$ is the applied force. This direct piezoelectric effect provides a new pathway for fabricating self-powered sensors. Therefore, sensors based on piezoelectric materials exhibit low-to-zero power consumption, ultrathin structure, high sensitivity, high flexibility and fast response time, which have attracted considerable research efforts recent years [16,17].

There are two basic working principles for piezoelectric sensors. One is the piezotronic effect, namely, for piezoelectric semiconductor materials such as ZnO, AlN and GaN, strain-/pressure-induced polarization charges at the interface modulate the local interfacial band structure, and eventually cause a current change on the piezoelectric semiconductor devices. Consequently, the current transport behavior reflects the mechanical deformation of piezoelectric materials under the applied force [18]. Accordingly, many piezotronic diodes and piezotronic transistors as strain/pressure sensors have been reported [19,20,21,22,23]. The first piezotronic field effect transistor based on a single ZnO nanowire was fabricated in 2006, which has a linear sensing ability when the applied force is around nanonewton [21]. He et al. fabricated a piezoelectric gated diode based on a single ZnO nanowire, which could be a device element for RAM applications under appropriate bending and voltage control [24]. The other is the piezoelectric effect. Due to the accumulation of the charge, the piezoelectric potential is generated at the upper and lower surfaces of the piezoelectric material, which can be expressed as:(3)$$U=\frac{Q}{C}=\frac{d_{mn}t}{\varepsilon_{r}\varepsilon_{0}A}\cdot F_{x}$$
where t is the thickness, $\varepsilon_{r}$ is the dielectric coefficient; $\varepsilon_{0}=8.854\times 10^{-12}{\mathrm{Fm}}^{-1}$ and A is the area of surface. The piezoelectric potential can generate pulse current in external circuits to convert mechanical energy into electrical energy, which can be made into generators at the nanometer-scale and collect the weak and irregular mechanical energy in the environment [25]. In addition, the change of potential directly reflects the external applied force, which is the basis for piezoelectric sensors. Accordingly, the piezoelectric material can be used to fabricate flexible piezoelectric sensors. Compared with traditional piezoresistive sensors, these piezoelectric sensors are self-powered, which directly convert the applied force into voltage signal. The sensitivity ($K_{u}$) of piezoelectric sensors can be expressed as:(4)$$K_{u}=\frac{dU}{dF_{x}}=\frac{d_{mn}t}{\varepsilon \varepsilon_{0}A}$$

This indicates that piezoelectric sensors have linear sensing ability, therefore hardly require extra complex signal process. Self-powered flexible piezoelectric sensors with linear sensing ability meet the growing demand in the flexible microelectronic system with small size and low power consumption. Hence, they are considered excellent prospects for sensing applications.

Lately, owing to high flexibility, fast response, high sensitivity, broad frequency bandwidth, high dynamic range, low cost and ease of processing, piezoelectric sensors have found a large number of applications in wearable electronics [26,27,28,29], medical equipment [30,31], structural health monitoring [32,33,34,35,36], machining process monitoring [37,38,39,40,41], structural parameter measurement [42,43,44,45], surface acoustic wave detection devices [46], hard disk drive [47,48] and microphones [49]. The development history of piezoelectric sensor applications in recent years is shown in Figure 1.

As in Figure 1, in 2009, a suspension was proposed, and a PVDF film sensor was attached to this suspension. Then, it was built into a real hard disk drive. The prototype is shown in Figure 1a. By comparing the power spectrum of disk vertical vibration measured by Laser Doppler Vibrometer (LDV) with that of PVDF film sensor output, disk flutter can be well detected at several frequencies: 1.4, 1.8, 2.4, 3 and 4 KHz. In structural health monitoring, machine failures monitoring and industrial process, compared with the MEMS accelerometers, acoustic emission sensors and traditional sensors, which suffer from mounting intrusiveness to the process, lowering of the dynamic stiffness of the machine tool system, limited bandwidth, high dependence on workpiece mass and geometry and high cost, organic piezoelectric materials such as PVDF have attracted wide interest due to their sufficient flexibility suitable for a specimen of irregular shape, lightweight, wide frequency bandwidth, high strain sensitivity and low cost. For example, in 2012, Ma et al. designed a sensor module that integrated an in situ data logging platform and a thin film PVDF strain sensor and mounted it on the tool shank for monitoring of the feed and transverse forces in the peripheral end milling process. However, these are still some challenges needed to be studied and solved in these fields including (1) attachment methods to achieve high robustness and withstand intense vibrations or large strain; (2) device structure design or material selection to obtain wide frequency bandwidth and high strain sensitivity; (3) wireless data acquisition and data transmission. In 2014, Chen et al. fabricated flexible and transparent ZnO thin film strain sensors on flexible glass substrates, which worked well under various applied strains up to ±3000 με and showed excellent linearity of resonant frequency with a sensitivity of $34{\;\mathrm{Hz}\mu \varepsilon }^{-1}$. This work demonstrates great potential applications of the surface acoustic wave (SAW) device based on strain sensors. In 2016, as shown in Figure 1b, Shin et al. fabricated a wearable and wireless pressure sensor based on ZnO nanoneedles and PVDF hybrid films, of which the lowest detectable pressure was 4 Pa. The heart rates monitored by the sensor were transmitted to the smart phone via the reduced graphene oxide (rGO) electrode-based Bluetooth antenna. In 2017, Wei et al. designed and fabricated a 1-D force sensor based on PVDF and macro-fiber composite (MFC) films with a sensitivity of $1.23{\;\mathrm{mVmN}}^{-1}$, a resolution of 0.80 mN and a measuring range of 100 mN. A prototype was fabricated and applied to crab egg embryo injection (Figure 1c). In addition, a performance assessment of macro-fiber composite (MFC) sensors were presented for measuring acoustic emission (AE) signals from partial discharges (PD) in power transformers filled with mineral oil [50] (Figure 1d). In 2018, Zhang et al. proposed a non-intrusive method to detect train wheel damage by adding a thin PVDF-based multilayered sensing device beneath the rail pad. The sensor based on PVDF showed outstanding linearity with a sensitivity of $72.2{\;\mathrm{mVkN}}^{-1}$. These reports show that piezoelectric sensors will have more and more potential applications in modern biologic medicine, wearable devices, structural health monitoring.

The research and development of high-performance sensors based on piezoelectric materials with high sensitivity, wide sensing range, fast response, flexibility and low-to-zero power consumption has a great significance for flexible microelectronic system. Hence, in this review, several common strategies to improve the performance of sensors based on piezoelectric materials are summarized, including material innovation, from widely investigated and used piezoelectric semiconductor materials, inorganic piezoceramic materials, organic piezoelectric polymers, nanocomposite materials, to emerging and promising molecular ferroelectric materials with both high piezoelectricity and flexibility; designing microstructures on the surface of the piezoelectric materials to improve the contact area of piezoelectric materials under the applied pressure; the addition of dopants such as chemical elements and graphene in conventional piezoelectric materials; and developing piezoelectric transistors based on piezotronic effect as piezoelectric sensors (Figure 2). Meanwhile, the principles of how these factors affect the performance of piezoelectric sensors are analyzed and research recommendations are made to develop high-performance flexible piezoelectric sensors.

## 2. Effective Methods to Fabricate High-Performance Piezoelectric Sensors

### 2.1. Seeking for Materials with High Piezoelectricity and Flexibility to Improve the Performance of Piezoelectric Sensors

Piezoelectric materials play a decisive role in determining the performance of piezoelectric sensors. The sensitivity and sensing ranges of piezoelectric sensors fabricated with different piezoelectric materials vary greatly. To date, ZnO, PZT, ${\mathrm{BaTiO}}_{3}$ and PVDF are the major functional piezoelectric materials that have been fully researched and widely applied (Figure 3) [51,52,53,54,55].

ZnO is the earliest studied piezoelectric semiconductive material, due to its relatively simple chemical composition and crystal structure and easy control of purity, size and surface morphology [56]. Meanwhile, the semiconducting, desirable piezoelectric property, biosafety and easy fabrication make ZnO a great candidate for piezoelectric sensors. ZnO nanowires can generate electrical signals in response to small mechanical force for the sake of its large aspect ratio. Therefore, the electrical response of a single ZnO nanowire under small forces was reported in 2006 [21]. As shown in Figure 2a, Zhao et al. fabricated a strain sensor based on a single ZnO nanowire. Compared with a commercial strain sensor, its gauge factor is 200 times higher (above 400). However, the strain sensor only withstand ultra-small mechanical strain on the order of $10^{-6}$ [57].

Considering that the single nanowire will break under large force and generates a relatively poor electrical response, in practical applications, it is essential to improve sensitivity and broaden sensing range of piezoelectric sensors by integrating large numbers of ZnO nanowires on a flexible platform. For example, the field-limited ZnO nanorods were grown on a flexible Kapton substrate by using a low temperature hydrothermal method, which was then packaged with PDMS to fabricate a one-piece flexible tactile sensor with good linearity and sensitivity of $403{\;\mathrm{mVmPa}}^{-1}$ at around 0.7 mPa [23]. Meanwhile, the diameter, height and density of the micropillars/microrods can influence the performance of piezoelectric sensors, including flexibility, sensitivity and sensing range [58], which deserves to be further researched by analyzing the coupling effects of the mechanical and piezoelectric properties under different diameters, heights and density. As shown in Figure 3a, a study-based ZnO nanowires touch pad was designed [59], which exhibited a sensitivity of $0.57{\;\mathrm{nAN}}^{-1}$ and a linear relationship between the applied force and the response current. In addition, semiconductive piezoelectric materials combined with piezotronic, ZnO for instance, can be used to fabricated amperometric piezoelectric sensors, which will be analyzed in detail in Section 2.4.

However, the relatively poor piezoelectric characteristics of ZnO lead to the low sensitivity of ZnO-based piezoelectric sensors. Therefore, alternative materials with higher piezoelectric coefficient need to be found to fabricate high-performance piezoelectric sensors. Lead zirconate titanate (PZT) is a solid solution of two materials including a ferroelectric lead titanate (${\mathrm{PbTiO}}_{3}$) and an anti-ferroelectric lead zirconate (${\mathrm{PbZrO}}_{3}$) [60]. It owns high piezoelectric coefficient, outstanding electromechanical coupling effect, high dielectric constants and good crystallinity [61,62,63]. PZT has been vastly researched and used for piezoelectric sensors. For instance, as shown in Figure 3b, a flexible semi-transparent sensor based on laterally aligned PZT nanowires and interdigital Pt/Ti electrodes was assembled using a simple spin-coating method, which exhibits high pressure sensitivity (0.14 ${\mathrm{VkPa}}^{-1}$) [64]. A PZT diaphragm strain sensor with diameter of 1.35 cm and thickness of 300 μm was fabricated, which generated a voltage magnitude of 7.3 mV at the maximum displacement of 250 μm and can sense load forces between 89 and 539 kPa [65]. Owing to the outstanding piezoelectric characteristics of PZT-based ceramic materials, these piezoelectric sensors exhibits higher performance, which makes them more competitive in microelectronics. However, PZT contains more than 60 percent lead (Pb) by weight, which has an adverse effect on human beings and ecosystems. Therefore, regulators around the world have begun severely restricting the use of lead [12,63]. Compared with PZT, lead-free piezoelectric materials not only have no side effect on environment, but also exhibit comparable or even superior piezoelectric properties [55,56]. Therefore, they have attracted increasing attention in recent years.

Lead-free piezoceramics like barium titanate (BT), bismuth potassium titanate (BKT), potassium sodium niobate (KNN), bismuth sodium titanate (BNT) and so on, without PbO in its composition, are expected to replace lead-based piezoceramics in the future due to their environmental friendliness and biocompatibility [63,66]. Although BNT–BKT lead-free piezoceramics have a high Curie temperature, they have many disadvantages that cannot be ignored, such as highly corrosive due to the presence of alkalis, low piezoelectric properties and volatile [63]. For KNN-based lead-free ceramics with a high piezoelectric coefficient, their drawbacks such as volatility of alkali components at high temperature and instability of KNN phase at high temperature make it difficult to fabricate [60]. Therefore, BKT, BNT and KNN materials have not been widely investigated and developed.

The piezoelectric coefficient ($d_{33}$) of pure ${\mathrm{BaTiO}}_{3}$ through solid state processing is $\sim 190{\;\mathrm{pCN}}^{-1}$ [67], which is higher than that of pure ZnO. Owing to the excellent ferroelectric properties, piezoelectric properties and biocompatibility, the performance of ${\mathrm{BaTiO}}_{3}$-based sensors have been further improved. For example, as shown in Figure 3c, Zhou et al. prepared vertically aligned ${\mathrm{BaTiO}}_{3}$ nanowires on a conductive substrate based on a two-step hydrothermal reaction, which exhibited a high piezoelectric response ($d_{33}=43\pm 2\;\mathrm{pm}/V$). Besides, ultra-long vertically aligned ${\mathrm{BaTiO}}_{3}$ nanowire arrays were fabricated by a two-step hydrothermal reaction, of which the diameter and length, respectively, were about 600–630 nm and 40 μm [68]. When applied a sinusoidal base acceleration of 0.25 g rms at lower resonant frequency (170 Hz), the device outputed a high peak to peak voltage (345 mV). In 2017, a self-powered wearable sensor based on ${\mathrm{BaTiO}}_{3}$/alginate spherical composite beads was fabricated through the ionotropic gelation (IG) method [69]. In addition, Wu et al. fabricated a piezoelectric strain sensor based on field effect transistors with a high sensitivity of $1.608\times 10^{7}N^{-1}$ [70].

Although ${\mathrm{BaTiO}}_{3}$ has many outstanding advantages such as outstanding ferroelectricity, piezoelectricity, nontoxicity and easy synthesis, the high polarization temperature and inherently poor flexibility make it difficult to satisfy the increasing need of flexible piezoelectric sensors. PVDF and its derived copolymers have fabulous properties such as high deformability, low permittivity, high thermal stability and outstanding chemical durability [71], which make them the most promising materials for piezoelectric sensors. Besides, their low acoustic impedance close to human tissues and water improves the competitiveness in the field of medical diagnosis, health monitoring and wearable devices [72]. As shown in Figure 3d, a sensor with sensitivity ($18.376{\;\mathrm{kPa}}^{-1}$ at ~100 Pa), fast response (15 ms), wide pressure range (0.002–10 kPa) and high durability (7500 cycles) was designed and fabricated by weaving PVDF electrospun yarns coated with Poly(3,4-ethylene dioxythiophene) (PEDOT) [73]. In 2016, a highly flexible self-powered sensor was fabricated through printed circuit board (PCB) and near-field electrospinning (NFES) technology, which provided a possible industrial preparation process for low-cost production [74]. Lu et al. adopted spin coating and mask etching to fabricate a flexible 4×4 PVDF strain sensor array with high sensitivity ($12{\;\mathrm{mVkPa}}^{-1}$) and fast recover time (<25 μs) [75]. However, compared with traditional piezoceramics like PZT and ${\mathrm{BaTiO}}_{3}$, the $d_{33}$ of PVDF is relatively lower, which limits the performance of PVDF-based piezoelectric sensors. However, the β−phase content, the strongest polar moment contributing to piezoelectricity of PVDF among all the crystalline phases (α, β, γ, δ and ε phases) [76], could be increased by doping [77,78] or improving processing technology [71,79] to enhance piezoelectricity of PVDF. Electrospinning, as the most straightforward and versatile technique to prepare PVDF nanofibers with large length to diameter ratio, has been widely studied and optimized [73,80,81,82]. For example, Zheng et al. increased the β−phase content by adjusting the technical parameters during electrospinning [83]. The technical parameters could be potentially modified by adding a low boiling point solvent like acetone, decreasing the flow rate of the tip, shortening the distance between the tip and the collector and decreasing the environment temperature [84,85]. However, the β−phase semi-crystalline PVDF with high piezoelectric coefficients cannot be produced in large scale due to the difficult synthesis [11], which still needs a great effort to find more facile, efficient and stable methods to improve the β−phase content.

In summary, flexibility and piezoelectricity of piezoelectric materials are the key factors that determine whether they could be widely used to fabricate piezoelectric sensors in the future. Nevertheless, for the most well-researched piezoelectric materials such as ZnO, PZT, ${\mathrm{BaTiO}}_{3}$ and PVDF, none of them have both high flexibility and good piezoelectricity. Although ZnO has semiconductor properties and piezoelectricity, its flexibility and piezoelectric properties are relatively poor. Similarly, inorganic ceramic materials with high ferroelectric and piezoelectric properties such as PZT and ${\mathrm{BaTiO}}_{3}$ have inherently poor flexibility. Moreover, PZT-based piezoelectric materials will be gradually abandoned due to its toxicity to environment and living things. Despite high flexibility, PVDF and its derived copolymers have poor piezoelectricity. With the increasing need for high sensing performance and flexibility, piezoelectric sensors need to be fabricated with highly piezoelectric and flexible materials. Therefore, new piezoelectric materials with high piezoelectricity and flexibility need to be explored and designed.

Composed of organic polymers of high flexibility and inorganic particles of high piezoelectricity, flexible nanocomposite materials provide a new solution to fabricate piezoelectric sensors with high sensing performance and flexibility. Hence, designing nanocomposite materials of high piezoelectricity and flexibility is another research hotspot in piezoelectric sensors field. The concept of piezoelectric ceramic/polymer composites was first proposed in 1978 [86]. Composite materials, the mixture of two or more materials, are formed by dispersing inorganic nanoparticles into an elastomeric matrix without a chemical reaction occurring between them [87]. Piezoelectric sensors based on composite materials are sensitive to the applied force due to high piezoelectricity of inorganic materials in composite materials. In addition, they show high conformality and flexibility owing to high elasticity of the elastomeric matrix in composite materials.

Besides high flexibility and piezoelectricity, composite materials have many other advantages such as eco-friendliness, cost-effectiveness and ease of large-scale fabrication. To date, the most commonly used inorganic materials are ZnO and $BaTiO_{3}$. The most used elastomeric matrixes are non-piezoelectric organic materials (i.e., polydimethylsiloxane (PDMS), polyvinyl alcohol (PVA), polystyrene (PS)) and piezoelectric organic materials such as PVDF and its derived polymers [58,88,89,90,91,92,93,94,95,96,97]. As shown in Figure 2b, Hu et al. improved sensing performance of the P(VDF−TrFE) nanofibers-based sensor by utilizing ${\mathrm{BaTiO}}_{3}$ NPs. The ${\mathrm{BaTiO}}_{3}$/P(VDF−TrFE) composite material exhibits the maximum β-phase crystallinity and piezoelectricity when adding 5 wt% ${\mathrm{BaTiO}}_{3}$ NPs. The sensor could detect the creep force of an ant (~1 mg). Figure 4 shows the micro-morphology, device design and sensing performance of piezoelectric sensors based on composite thin film materials.

As shown in Figure 4a, a high-strain sensor based on ZnO NWs and polystyrene nanofiber (PSNF) was fabricated by electrospinning [91], which can measure and withstand strains up to ~50%. However, due to the poor piezoelectricity of ZnO nanowire, its gauge factor is only ~116. To further improve sensing performance of piezoelectric sensors, inorganic materials with higher piezoelectricity need to be used. Therefore, Bian et al. adopted dip-coating method to fabricate a biomimetic force sensor by combining ${\mathrm{BaTiO}}_{3}$ nanoparticles with PDMS based on 3D polypyrrole (PPy) network (Figure 4b), which showed an excellent stretchability of up to 310%. Due to high piezoelectricity of ${\mathrm{BaTiO}}_{3}$ and three-dimensional porous network, the sensor could detect acceleration with sensitivity of $1.03{\;\mu \mathrm{As}}^{2}m^{-1}$ in a range from $1.7{\;\mathrm{ms}}^{-2}$ to $16.2{\;\mathrm{ms}}^{-2}$, static pressure with sensitivity of $12.6{\;\mathrm{pFkPa}}^{-1}$ in a range from 0.6 kPa to 5 kPa and dynamic pressure with sensitivity of $57.58{\;\mu \mathrm{AMPa}}^{-1}$ in a range from 31.25 kPa to 187.5 kPa [95]. In these composite thin film materials, the non-piezoelectric polymer materials such as PDMS and PS only work as supporting cross-linker materials to hold inorganic nanoparticles to form a non-directional polymer structure, which cannot improve piezoelectric properties or the electrical energy response of composite materials. In contrast, piezoelectric organic polymers such as PVDF and P(VDF-TrFE) act as not only a support layer, but also a piezoelectric active layer to enhance piezoelectric properties of composite materials. Moreover, the addition of inorganic materials can improve the β−phase contents in piezoelectric organic polymers [89]. The synergistic coupling effect between inorganic materials and piezoelectric organic polymer materials makes piezoelectric polymers promising materials for composite materials. For example, a flexible self-powered sensor (Figure 4c) on the base of the cowpea-structured PVDF/ZnO nanofibers with an fantastic bending sensitivity ($4.4{\;\mathrm{mVdeg}}^{-1}$), a wide range (from 44 degrees to 122 degrees), and ultrahigh pressing sensitivity $(0.33{\;\mathrm{VkPa}}^{-1}$) was fabricated, which was used for remote control of gestures in human-machine interactive system [54]. One of the major problems in the preparation of composite materials is to make inorganic nanoparticles homogeneously dispersed in organic polymers. However, traditional methods including hot-embossing, spin-coating and solvent-casting lead to heterogeneous dispersion of inorganic nanoparticles in polymer materials owing to the higher density of inorganic nanoparticles compared with polymer materials and agglomeration during fabrication process, which weakens mechanical and functional properties of composite materials. Therefore, Kim et al. adopted filament extrusion and 3D printing to fabricate homogeneous composite materials composed of ${\mathrm{BaTiO}}_{3}$ nanoparticles and PVDF [98]. The result showed that the piezoelectric response of nanocomposites fabricated with 3D printing exhibited 3 times higher than that of nanocomposites fabricated with solvent-casting.

In addition to investigating and exploring nanocomposite materials, researchers have been seeking flexible materials with both high piezoelectricity and flexibility. Due to their light weight, flexibility, low acoustical impedance, ease of low-temperature processing, structural tenability and biocompatibility, molecular ferroelectrics are potential promising materials for the exploration of next generation piezoelectric devices [99]. To date, various molecular ferroelectrics have been designed through different screen strategies. As shown in Figure 5a, Liao et al. found and researched a molecular perovskite ${\left(\mathrm{TMFM}\right)}_{x}{\left(\mathrm{TMCM}\right)}_{1-x}{\mathrm{CdCl}}_{3}$(0≤x≤1) (TMFM, trimethylfluoromethyl ammonium; TMCM, trimethylchloromethyl ammonium) solid solution [100]. Different piezoelectric materials can be synthesized by changing the value of x. When x equals 0.26, the synthesized piezoelectric material has a $d_{33}$ of ~1540 pC/N, which is two times than that of PZT. In addition, Ye et al. designed the family of metal-free ${\mathrm{ABX}}_{3}$ -type (where A is a divalent organic cation and X is Cl, Br or I) 3D perovskites (Figure 5b), among which $\mathrm{MDABCO}\;\left(N-\mathrm{methyl}-N’-\mathrm{diazabicyclo}\;\left[2.2.2\right]\;\mathrm{octonium}\right)-{\mathrm{NH}}_{4}I_{3}$ has a large spontaneous polarization ($P_{s}=22\;{\mu \mathrm{Ccm}}^{-2}$) and a high phase transition temperature ($T_{0}$ = 448 K) [101].

Unlike inorganic ferroelectrics whose polarization can be switched among multiple directions easily in each grain, the random grains of uniaxial molecular ferroelectrics are only polarized between two opposite polarization directions under the electric field, which leads to a weak macroscopic polarization and low piezoelectricity [102]. Therefore, multiaxial molecular ferroelectrics are proposed and researched. As shown in Figure 5c, You et al. discovered an organic-inorganic perovskite ferroelectric material of $\mathrm{TMCM}-{\mathrm{MnCl}}_{3}$ with a $d_{33}$ value of 190 pC/N and a high transition temperature of 406 K. At the same year, a molecular ferroelectric thin film [103] based on $\mathrm{TMBM}-{\mathrm{MnBr}}_{3}$ (${\mathrm{Me}}_{3}{\mathrm{NCH}}_{2}{\mathrm{BrMnBr}}_{3}$) was fabricated through dip-coating, which has a large piezoelectric coefficient ($d_{33}\sim 112\mathrm{pC}/N$) (the crystal structure model of $\mathrm{TMBM}-{\mathrm{MnBr}}_{3}$ is shown in Figure 5d).

Despite the high piezoelectricity and flexibility, strict design and screening strategies, and manufacturing conditions limit the development of molecular ferroelectrics [102]. Therefore, there are still many research and engineering work to be conducted. On the other hand, although nanocomposite materials have been researched widely, many challenges need to be tackled such as the refined control of surface roughness, developing composite materials with higher piezoelectricity and the best optimization of weight ratio of inorganic materials and organic polymer materials [104].

### 2.2. Designing Micro-Morphologies/Microstructures on the Surface of Materials to Improve the Performance of Piezoelectric Sensors

Apart from high-performance piezoelectric materials, precisely controlling the micro-morphology on the surface of materials can enhance the performance of sensors as well, which usually adopts different preparation technologies under rational reaction temperature. Micro-morphologies/microstructures on the surface of materials could increase contact areas and redistribute the applied force uniformly. For example, a self-powered piezoelectric sensor (Figure 2c) based on vertically aligned $P\left(\mathrm{VDF}-\mathrm{TrFE}\right)/{\mathrm{BaTiO}}_{3}$ micropillars was fabricated through nanoimprinting process, which could be used for detecting air pressure working in a noncontact mode [58]. The performance of piezoelectric sensors with different micro-morphologies is summarized in Table 1.

To detect the larger strain, a laterally aligned, patterned and highly stretchable polyaniline/poly (vinylidene fluoride) (PANI/PVDF) strain sensor was developed by electrospinning and in situ polymerization [82]. Compared with common nonwoven PANI/PVDF mat, it could detect a strain up to 110% and exhibited an excellent linear response to a wide strain range (0–85%). Besides, various sensors based on laterally aligned nanofibers were fabricated via electrospinning [105,106], which exhibit excellent stretchability and sensing performance. However, the response to normal force of these sensors is nonideal due to its planarization structure. Therefore, vertically aligned structures were designed to detect normal force efficiently. In vertically aligned structures, traditional one-dimensional nanostructures include nanowires, nanorods, nanopillars, nanotubes and nanoribbons. Besides, Chen et al. fabricated a vertically well-aligned P(VDF-TrFE) nanowires-based piezoelectric sensor with a high sensitivity (458.2 mV$N^{-1}$) [107], which were applied as wearable pressure sensors to detect some weak human activities such as breath, low magnitude sound wave and heartbeat pulse. To survive, living things in nature has evolved their micro- and nano-structures to efficiently response to environmental changes. Therefore, researches sought inspiration from those unique structures of biologic systems to further improve the performance of piezoelectric sensors. Inspired by the structure of human fingertips, Park et al. fabricated an interlocked and hierarchical microstructured ferroelectric sensor based on reduced graphene (rGO) and PVDF composite films [108], which could detect various stimuli ranging from vibration, static and dynamic pressure to temperature. By adopting this interlocked structure, small spots between interlocked microdomes could concentrate and amplify stress, which then greatly deforms the interlocked microdomes and increases contact areas. Consequently, this kind of sensors are more sensitive to small pressure.

Designing different micro-morphologies on material surface could increase contact area and distribute the applied force more uniformly to improve the performance of piezoelectric sensors. However, due to a nominal enhancement of piezoelectric properties from micro-morphologies, the performance of sensors based on such materials is restricted by their inherently piezoelectricity.

### 2.3. Adding Dopants to Improve the Performance of Piezoelectric Sensors

To date, piezoelectric sensors based on ZnO, PZT, ${\mathrm{BaTiO}}_{3}$ and PVDF exhibit good performance, but they are restricted by their piezoelectric properties. Meanwhile, designing different micro-morphologies on material surface is not essential to improve the piezoelectric properties. In addition, although a series of achievements were reported in molecular ferroelectrics, there are many theoretical and engineering problems need to be researched and optimized before its large-scale applications. On the other hand, composite materials are produced by dispersing inorganic nanoparticles into an elastomeric matrix without a chemical reaction occurring between them, of which piezoelectric properties are mainly decided by inorganic piezoelectric materials.

To further improve the performance of piezoelectric sensors based on ZnO, PZT, ${\mathrm{BaTiO}}_{3}$ and PVDF, various dopants (i.e., metallic elements, graphene) were researched and added into piezoelectric materials to increase the piezoelectric coefficient ($d_{33}$) [111,112,113,114,115]. For example, as shown in Figure 2d, graphene was used to induce the structural units of −${\mathrm{CH}}_{2}$− and −${\mathrm{CF}}_{2}$− of PVDF nanofiber chains to arrange directionally during phase separation to enrich the electroactive phase. The flexible fabrics-based piezoelectric sensor coated with PVDF/graphene exhibited high sensitivity of $34{\;\mathrm{VN}}^{-1}$ and low detecting limit of 0.6 mN [116]. Piezoelectric sensors based on dopant-modified ZnO, PZT and PVDF are shown in Figure 6.

Adding different chemical dopants into piezoelectric materials could modify their piezoelectric coefficients and dielectric constants, therefore improve their piezoelectric properties. Due to the semiconductor properties, the doping methods for ZnO-based piezoelectric materials are classified into n-doping and p-doping. For n-doping, dopants-induced strain of the lattice along the polar c-axis can improves the piezoelectric coefficient of ZnO crystal. Nevertheless, the addition of dopants with large ionic radius or high doping concentration would lead to many lattice defects, which increases resistance to the flow of electrons and decreases the performance of ZnO-based piezoelectric sensors. For p-doping, the dopant-generated shielding effect can be reduced in ZnO nanowires [11]. To date, dopants that have been researched to improve the piezoelectric properties of ZnO-based piezoelectric materials include halogen ions($F^{1-},\;{\mathrm{Cl}}^{1-},\;{\mathrm{Br}}^{1-},\;I^{1-}$) [117,118], metallic elements ($Y^{3+},{\mathrm{Fe}}^{3+},{\mathrm{Cr}}^{3+},V^{5+},{\mathrm{La}}^{3+},{\mathrm{Mn}}^{2+},{\mathrm{Mg}}^{2+},{\mathrm{Gd}}^{3+},{\mathrm{Eu}}^{3+},{\mathrm{Ce}}^{3+},{\mathrm{Cu}}^{3+},$ Fe, Ag, Li, V, Ga) [111,119,120,121,122,123,124,125,126,127,128,129,130,131,132]. For halogen-doping, the radius of $F^{-1}$ is slightly smaller than that of $O^{2-}$, which induces a compressive strain within the crystal structure, and then, decreases the piezoelectric performance. In contrast, the radius of ${\mathrm{Cl}}^{1-},{\;\mathrm{Br}}^{1-}{\;\mathrm{and}\;I}^{1-}$ are larger than that of $O^{2-}$, which expand the strain along the c-axis of the lattice, then increase the piezoelectric performance. However, the radius of $I^{1-}$ is much larger than that of $O^{2-}$, which leads to the flaw and distortion of the ZnO lattice and enhancement of the screening effect generated by free electrons, and then, leads to a decrease in piezoelectric performance [119]. Generally, owing to polarization rotation generated under external electric fields, the ZnO nanowires doped by metal ions exhibit higher piezoelectric coefficients compared with pure ZnO nanowires. From the microscopic origin, the ${\mathrm{Zn}}^{2+}$ ion on the ZnO crystal site would be substituted by the doped metal ion ($M^{n+}$), leading to the form of an M–O bond. It is noted that the radius of metallic ion influences the piezoelectric properties. When the size of the doped metal ion is small, substitution of $M^{n+}$ ion for ${\mathrm{Zn}}^{2+}$ results in easier rotation of non-collinear M–O bonds along the c-axis under the applied field, which produces large piezoelectric displacement and corresponding piezoelectric response enhancement. When the size of the doped ion is large, it is difficult to rotate the non-collinear M–O bond, leading to a decrease in the piezoelectric response of the ZnO. That is, doping ZnO with a small ion produces enhanced piezoelectric response, whereas doping ZnO with a larger ion results in decreased piezoelectric response. However, when the size of the doped ion is large, the orientation of the ZnO nanowires along the c-axis will be enhanced, and the lattice constant along c-axis in the ZnO crystals will decrease. Therefore, the synergistic effect between enhanced orientation and decreased lattice constant will enhance the piezoelectric response of ZnO nanowires [11]. For example, as shown in Figure 6a, a Li-doped ZnO in-plane strain mapping sensor was fabricated by radio frequency (RF) sputtering. The X-ray diffraction patterns of the un-doped and Li-doped ZnO thin films demonstrated that the un-doped film has a (0002) peak corresponding to ZnO single crystal at 34.20 degrees, which reveals that the c-axis of ZnO thin films is perpendicular to the substrate. Meanwhile, the corresponding peak of Li-doped ZnO film is at 34.24 degrees. When the Li substitutes the Zn site, the lattice constant of ZnO will reduce a little bit, which induces this tiny peak shift. Furthermore, Song et al. conducted the test of carrier type and conductivity by the hall effect, which demonstrated that doped ZnO film is still an n-type semiconductor. The in-plane strain mapping sensor achieved a high gauge factor (~199) through adapting nanowire arrays and doping Li element [115].

Overall, chemical doping could be regarded as practical routine for enhancing performance of piezoelectric sensors on conditions of reasonably controlling the ionic size of dopants and doping concentration.

The piezoelectric properties of inorganic ceramic PZT- and ${\mathrm{BaTiO}}_{3}$-based piezoelectric materials could also be improved by doping chemical elements. For example, due to the coexistence of polymorphic phase boundaries in Zr/${\mathrm{BaTiO}}_{3}$ hybrid piezoelectric thin films [133], the piezoelectric coefficients of ${\mathrm{BaTiO}}_{3}$ thin films doped with Zr was enhanced greatly. The ${\mathrm{Ti}}^{4+}$ were substituted by ${\mathrm{Zr}}^{4+}$ and ${\mathrm{BaTi}}_{\left(1-x\right)}{\mathrm{Zr}}_{x}O_{3}$ was formed, which led to the increasement of the interplanar spacing of crystals and the shift of the charge center inside the crystal. These two factors synergistically increased the piezoelectric coefficients (285–290 ${\mathrm{pCcm}}^{-2}$) of ${\mathrm{BaTi}}_{\left(1-x\right)}{\mathrm{Zr}}_{x}O_{3}$(0.02≤x≤0.08) hybrid piezoelectric materials [134]. In 2017, Zhuang et al. fabricated a flexible tactile sensor based on Ce-doped ${\mathrm{BaTiO}}_{3}$ nanofibers and analyzed the doping mechanism and working process [135]. In addition, a nanocomposite film force sensor based on La modified PZT/paint was developed with brushing technique [136], which exhibited high performance with the output voltage of 51.7 mV and power of 0.38 μW when working in the -31- piezoelectric mode (Figure 6b).

Not only the inorganic piezoelectric materials, but also organic piezoelectric materials like PVDF and its derived copolymers could be doped with chemical elements such as Ag,${\;\mathrm{Ce}}^{3+}$ and ${\mathrm{Eu}}^{3+}$ to improve the piezoelectric properties [112,113,137]. For example, the addition of ${\mathrm{Eu}}^{3+}$ could prevent external polarization conditions from damaging the β−phase of P(VDF-HFP) and induce self-polarization of its electroactive β-phase, which maintains the β−phase content and leads to higher piezoelectric coefficients compared to the pure P(VDF-HFP) [112]. As shown in Figure 6c, the ${\mathrm{Ce}}^{3+}$ doped PVDF/graphene nanofibers for nanopressure sensor and acoustic nanogenerator were developed via in situ poled fabrication strategy [113], which exhibited a well-behaved, linear response to the applied pressure and could detect a low pressure of 2 Pa. In addition, for PVDF organic piezoelectric materials, reduced graphene or graphene are sometimes used to enhance piezoelectricity by transforming the PVDF phase from the nonpolar α-phase to the polar β-phase [138,139]. However, too high doping concentration of reduced graphene would result in the decrease of piezoelectric performance. This behavior can be attributed to the enhanced screening of piezoelectric polarization with the increase of film conductivity due to higher reduced graphene doping [108,140].

Although the performance of piezoelectric sensors could be improved by elaborately choosing the radius of dopants and dispersing dopants uniformly in matrix, all the reports only makes qualitative analysis from semiconductor physics (lattice), mechanism (stress redistribution), electricity (dielectric constant). Therefore, the detailed principle behind doping should be further researched and analyzed quantitatively to establish effective guidelines.

### 2.4. Utilizing Piezotronics to Fabricate Strain-/Pressure-Driven Gated-Controlled Sensing Transistors to Improve the Performance of Piezoelectric Sensors

The coupling effect of piezoelectricity and semiconductor properties of piezoelectric semiconductor materials brings about some impressive characteristics and provides theoretical basis for piezotronics [19,20,141]. In 2006 and 2007, Wang et al. proposed the concept of piezotronics, which aimed to study the coupling mechanism between piezoelectricity and conductivity of piezoelectric semiconductor materials [142]. Since then, this field has attracted increasing research interests and many piezoelectric sensors based on piezoelectric transistors have been reported.

When an external force is applied to the piezoelectric semiconductor materials, piezoelectric potential is created due to the strain-/pressure-induced polarization charges. If the piezoelectric material is connected to a metal (metal-semiconductor (M-S) contact formed) or other semiconductor (p-n junction formed), polarization charges at interface modulate the local interfacial band structure and then influence the concentration/distribution of free carriers and interfacial electronic charged states in a M-S contact or p-n junction. Consequently, the current in piezoelectric semiconductor devices is changed [143,144].

In recent years, many piezoelectric sensors based on piezotronic transistors by replacing the external gating voltage with the piezoelectric polarization have been reported, in which the current of transistors is modulated by the applied strain on piezoelectric semiconductor materials. As shown in Figure 7a, a strain sensor based on individual ZnO piezoelectric fine-wires was fabricated with a high gauge factor of 1250 at a strain of 0.99%, which is much higher than that of sensors discussed above [145]. Although many techniques have been reported to synthesize one-dimensional semiconductor nanostructures [146,147,148,149], they are either cumbersome or incompatible with state-of-art microfabrication technologies. Moreover, the single nanowire cannot withstand large force, which hinders their applications. Therefore, piezotronic thin films were researched and fabricated by the state-of-art microfabrication technologies and overcame the limitations existing in 1D nanostructures. For example, a strain sensor based on In-doped ZnO nanobelts was developed by the chemical vapor deposition (CVD) technique, which exhibited high gauge factor of 4036 (compressive strain) and 135 (tensile strain) [150]. As shown in Figure 7b, a strain sensor based on ZnO-paper nanocomposite materials was fabricated to monitor the real-time current response under both static and dynamic mechanical loading. Besides, a micro-strain sensor based on ZnO microwire network structure was developed under low temperature, of which the highest gauge factor was ~900 [151]. However, c-axes of crystal in thin films cannot be aligned well, which reduces the macroscopic piezoelectricity of piezotronic thin film and decreases the performance of piezoelectric sensors. Meanwhile, the response of the piezotronic thin films to normal force is nonideal due to its plane structure. Therefore, vertically well-aligned ZnO nanowires were fabricated to detect small normal pressure. For example, a device based on individually contacted vertical piezoelectric nanowires was fabricated by Ultraviolet (UV) and e-beam lithography techniques [152], which exhibited a high linearity with flow rate (Figure 7c). In addition, Liao et al. reported a pressure sensor based on vertically well-aligned ZnO nanorods [153] (Figure 2e). Due to the electron-tunneling modulation effect of MgO nanolayer, the sensor had a “on/off” ratio up to $10^{5}$ and exhibited a high sensitivity ($7.1\times 10^{4}\;{\mathrm{gf}}^{-1}$) and a fast response time (128 ms). As shown in Figure 7d, a transistor with P(VDF-TrFE) as active materials and graphene as gate electrodes was fabricated to form an active matrix strain sensor, which exhibited a high gauge factor (389) and a low detectable strain (0.008%) [154].

Different from Ohmic-contacted sensors, of which the sensitivity is decided by bulk channel modulation under strain or pressure, the sensitivity of piezoelectric sensors based on Schottky contact is mainly decided by tuning local SBH at interface. Therefore, how to choose piezoelectric semiconductor and metal whose SBH at interface is highly sensitive to strain are still challenges in the future. In addition, various piezoelectric semiconductor materials with more excellent piezoelectricity are waiting to be investigated and developed.

## 3. Conclusions and Outlook

### 3.1. Conclusions

Due to the excellent perceptual mechanism based on piezoelectric effect, piezoelectric materials are made into ultra-thin, highly sensitive and fast response flexible sensors, which can be applied in robotics and prosthetics, wearable electronics, intelligent and artificial devices, modern medicine and structural health monitoring. However, the performance of flexible sensors based on piezoelectric materials is not perfect so there is still room optimization. Here, several strategies that could improve the performance of piezoelectric sensors are summarized: (1) material innovation: from widely researched and used piezoelectric semiconductor materials, inorganic piezoelectric ceramic materials, organic piezoelectric polymer, nanocomposite materials, to emerging and promising molecular ferroelectric materials showing high piezoelectricity and flexibility. (2) designing microstructures on the surface of the piezoelectric materials to enlarge the contact area of piezoelectric materials under the applied pressure. (3) the addition of dopants such as chemical elements and graphene in conventional piezoelectric materials. (4) developing piezoelectric transistors based on piezotronic effect as piezoelectric sensors. However, each method is imperfect and has its own drawbacks. The method of seeking for new materials has long development cycle, high cost and difficulty in engineering realization. For designing micro-morphologies/microstructures on material surface, its improvement on performance is limited since it only changes the contact area and dispersion degree of the applied pressure and does not improve the piezoelectricity of piezoelectric materials. Moreover, it required more complex preparation processes. The effective way to choose rational doping ratio and make dopants uniformly dispersed in piezoelectric materials is still not perfectly solved. In addition, the possible principle behind doping should be further researched and analyzed quantitatively to establish practicable guidelines. With regard to strain-/pressure-driven gated-controlled sensing transistors based on piezotronic effect, the benefits of self-supplying power and linear response are lost. Therefore, there are still some unsolved problems in fabricating piezoelectric sensors with high sensitivity, low-to-zero power consumption, high flexibility, wide sensing range, fast response and high durability.

Undeniably, each method mentioned above has its own special benefits. To fully utilize the advantages of each method, several methods are used together to fabricate piezoelectric sensors to further improve the performance of piezoelectric sensors. For example, micro-morphologies/microstructures are designed and prepared on the surface of doped piezoelectric materials. Moreover, individual device plays a more and more important role when embedded in a system. For instance, connected to a wireless transmission module, the flexible piezoelectric sensor can detect and send the arterial pressure signals to the smart phone to construct a real-time monitoring system [155]. The circuit design is also a good choice to enhance the performance of piezoelectric sensors. For example, a charge amplifier is connected to a piezoelectric sensor to convert charge accumulating on the opposite surfaces of the piezoelectric sensor into voltage. The output voltage could be expressed as:(5)$$V_{\mathrm{out}}=\frac{1}{C_{F}}\;\int -Idt=-\frac{Q}{C_{F}},$$
where $C_{F}$ is the feedback capacitance. This expression indicates that the gain only depends on the feedback capacitance $C_{F}$ and is not affected by the cable capacitance or the internal capacitance of the piezoelectric sensor. In this case, the sensitivity could be expressed as:(6)$$K_{u}=\frac{dV_{out}}{dF_{x}}=-d_{mn}\frac{1}{C_{F}}$$

It is noted that the charge amplifier should have high input impedance to ensure that the small amount of charge generated by the piezoelectric sensor is not lost through leakage.

### 3.2. Outlook

With artificial intelligence and cloud computing gradually developing into mature technologies, the Internet of things (IOT) era is coming. Meanwhile, wearable devices based on flexible electronics will be the mainstream of next-generation portable devices. As the only functional device for data acquisition, sensors are playing a more and more important role in IOT era. Therefore, developing high-performance sensors with great sensitivity, low-to-zero power consumption and low cost is a big challenge in the development of the next generation wearable devices and electronic skin. Further advances in piezoelectric sensors will involve: (1) high-performance piezoelectric sensors, (2) preparation technology improvements, (3) application-specific designs, (4) integrated technology with other electronic devices.

The methods for designing and preparing high-performance piezoelectric sensors are the most critical part of future research orientation, which involves materials innovation, structure innovation, material optimization and principle innovation. Many reported piezoelectric sensors have met the requirement of accuracy in most applications. However, there are many other problems needed to be solved. Most of preparation processes reported are carried out in lab conditions. Therefore, due to the uncertainty about complexity, repeatability and stability, mass production of piezoelectric sensors under low cost condition has not been extensively attempted in factory conditions, which is one of the biggest stumbling blocks to their practical applications. The fabrication technology like spin-coating, inkjet printing and screen printing may hold hope to lower the cost and realize massive production of piezoelectric sensors. In addition, for specific applications, some other properties like adaptability, durability, shape and size need to be customized. For instance, to monitor personal health or human posture, these wearable and implantable electronic devices need to be tightly integrated in clothing or human skin. Therefore, piezoelectric sensors are required to have small size, high flexibility and adaptability. In addition, single device would play a more important role when embedded in a system. Piezoelectric sensors with low-to-zero power consumption meets the requirements of future flexible microelectronic systems with small size and low consumption. Therefore, researching and developing integrated technology of piezoelectric sensors with other electronic devices to achieve complete and complicate functions is another research hotpot.

## Figures and Tables

**Figure 1 sensors-20-02820-f001:**
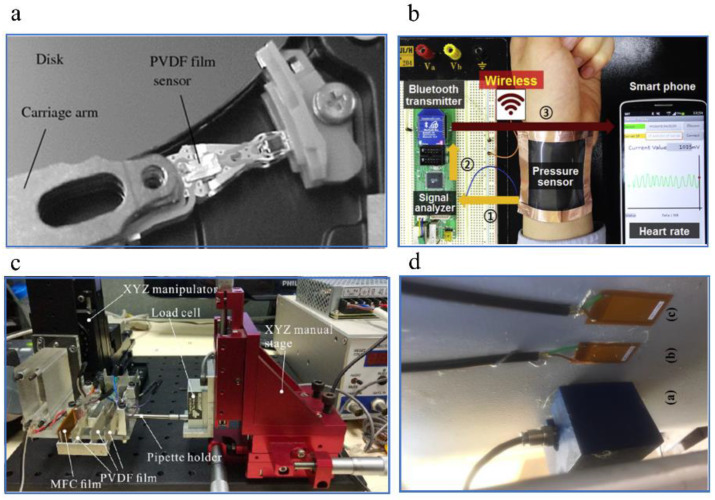
Development of piezoelectric sensor applications. (**a**) A polyvinylidene fluoride (PVDF) film sensor was used in hard disk drivers in 2009. Copyright from Springer Nature. (**b**) A sensor based on ZnO nanoneedles and PVDF hybrid films was used as heart rate monitor in 2016. Copyright from Elsevier. (**c**) A 1-D force sensor based on PVDF and macro-fiber composite (MFC) films was applied to crab egg embryo injection in 2017. Copyright from IEEE. (**d**) Macro-fiber composite (MFC) sensors were presented for measuring acoustic emission signals in 2017. Copyright from IEEE.

**Figure 2 sensors-20-02820-f002:**
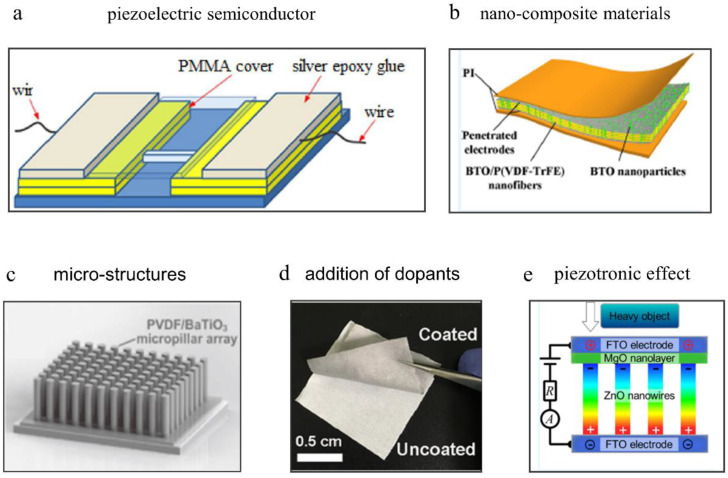
Different strategies facilitating high-performance flexible piezoelectric sensors. (**a**) material innovation: a ZnO-based piezoelectric sensor. Copyright from Springer Nature. (**b**) material innovation: a piezoelectric sensor based on nanocomposite materials. Copyright from ACS. (**c**) Designing microstructures: A piezoelectric sensor based on micropillar array. Copyright from John Wiley and Sons. (**d**) addition of dopants: a piezoelectric sensor based on phase-separation-induced PVDF/graphene coating on fabrics. Copyright from ACS (**e**) piezoelectric sensors based on piezotronic effect: a pressure sensor based on vertically well-aligned ZnO nanorods. Copyright from ACS.

**Figure 3 sensors-20-02820-f003:**
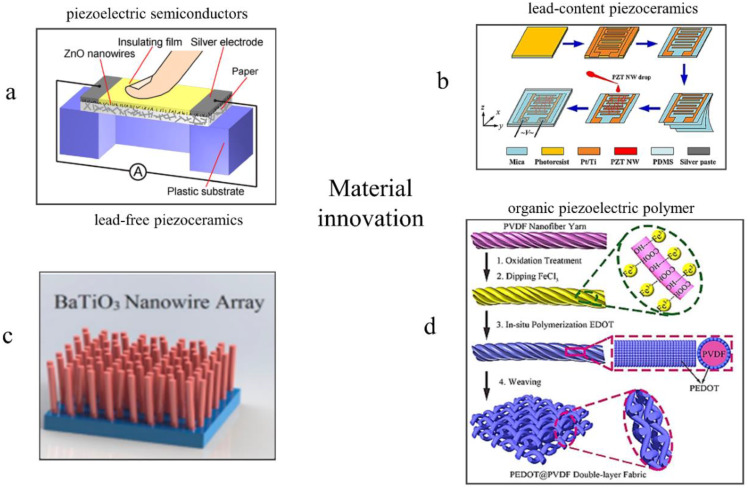
Material innovation: piezoelectric semiconductors, inorganic piezoceramics and organic piezoelectric polymer. (**a**) structure of a sensor based on ZnO. Copyright from ACS. (**b**) structure of a sensor based on ${\mathrm{Pb}}_{x}{\mathrm{Zr}}_{1-x}{\mathrm{TiO}}_{3}$ (PZT). Copyright from ACS. (**c**) structure of a sensor based on ${\mathrm{BaTiO}}_{3}$. Copyright from ACS. (**d**) structure of a sensor based on PVDF. Copyright from Springer Nature.

**Figure 4 sensors-20-02820-f004:**
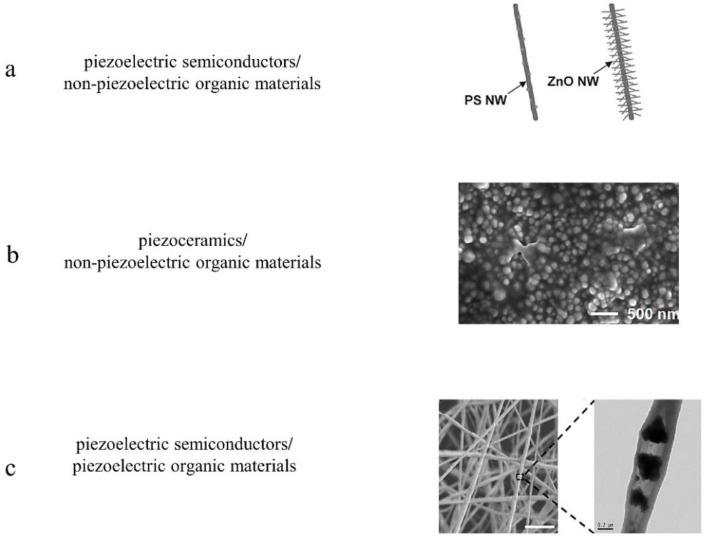
Material innovation: nanocomposite materials. (**a**) SEM image of a sensor based on ZnO/PS NWs. Copyright from John Wiley and Sons. (**b**) SEM image of a sensor based on ${\mathrm{BaTiO}}_{3}$/PDMS. Copyright from Elsevier. (**c**) SEM image of a sensor based on ZnO/PVDF. Copyright from Elsevier.

**Figure 5 sensors-20-02820-f005:**
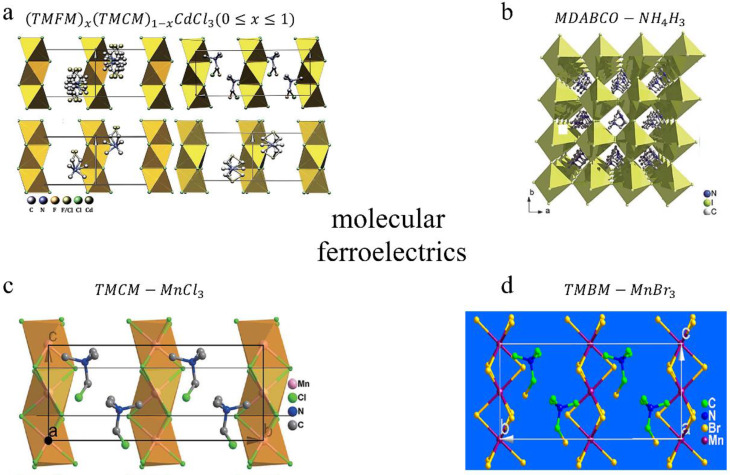
Material innovation: molecular ferroelectrics. (**a**) crystal structure of ${\left(\mathrm{TMFM}\right)}_{x}{\left(\mathrm{TMCM}\right)}_{1-x}{\mathrm{CdCl}}_{3}$(0≤x≤1). Copyright from The American Association for the Advancement of Science. (**b**) crystal structure of MDABCO–${\mathrm{NH}}_{4}H_{3}$. Copyright from The American Association for the Advancement of Science. (**c**) the crystal structure of
$\mathrm{TMCM}-{\mathrm{MnCl}}_{3}$. Copyright from The American Association for the Advancement of Science. (**d**) crystal structure of $\mathrm{TMBM}-{\mathrm{MnBr}}_{3}$. Copyright from ACS.

**Figure 6 sensors-20-02820-f006:**
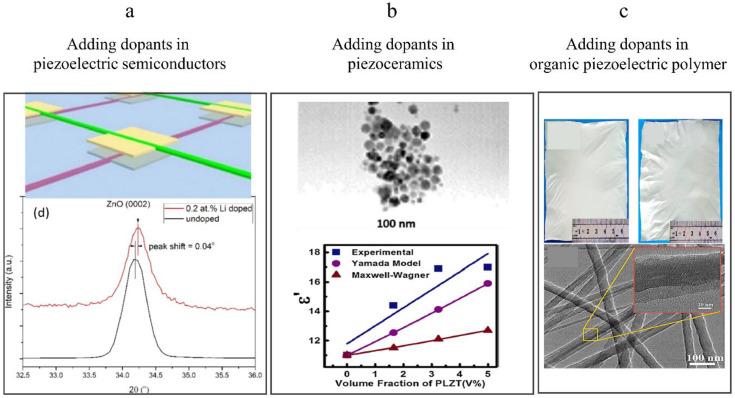
Addition of dopants. (**a**)Adding dopants in ZnO: structural schematic diagram (upper part) and XRD spectrum (lower part). Copyright from Elsevier. (**b**) Adding dopants in PZT: SEM image (upper part) and dielectric constants (lower part). Copyright from Springer Nature. (**c**) Adding dopants in PVDF: structural schematic diagram (upper part) and SEM image (lower part). Copyright from ACS.

**Figure 7 sensors-20-02820-f007:**
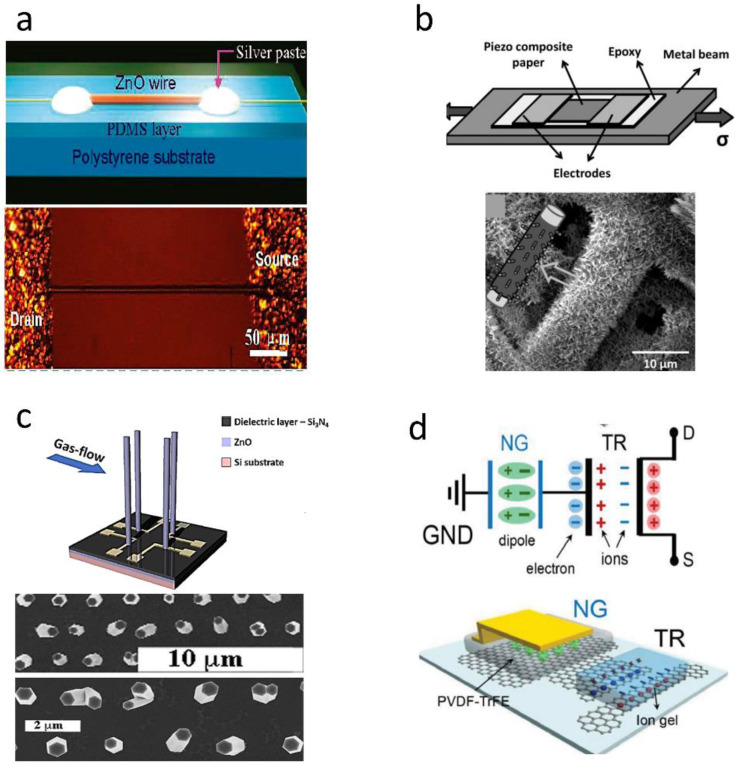
Sensors based on piezotronics effect. (**a**) strain sensor based on individual ZnO piezoelectric fine-wires: structural schematic diagram (upper part) and SEM image (lower part). Copyright from ACS. (**b**) strain sensor based on ZnO-paper: structural schematic diagram (upper part) and SEM image (lower part). Copyright from John Wiley and Sons. (**c**) sensor based on individually contacted vertical piezoelectric nanowires: structural schematic diagram (upper part) and SEM image (lower part). Copyright from Elsevier. (**d**) transistor with P(VDF-TrFE) as active materials and graphene as gate electrodes: circuit diagram (upper part) and structural schematic diagram (lower part). Copyright from John Wiley and Sons.

**Table 1 sensors-20-02820-t001:** Comparison of piezoelectric sensors with different microstructures.

Micro-Structures	Sensitivity	Detect Limit	Sensing Range	Response Time	Stability (Cycles)	Ref.
Laterally aligned P(VDF-TrFE) nanofibers	$1.1{\;\mathrm{kPa}}^{-1}$	0.1 Pa	0.4–2 kPa	–	–	[109]
PANI/PVDF nanofiber webs	1.84 ${\mathrm{kPa}}^{-1}$	–	0–110% (strain)	–	10,000	[82]
P(VDF-TrFE)/graphene nanofiber webs	$15.6{\;\mathrm{kPa}}^{-1}$	1.2 Pa	–	5 ms	100,000	[81]
Vertically aligned P(VDF-TrFE) nanowires	$0.046{\;\mathrm{kPa}}^{-1}$	–	–	–	36,000	[107]
Vertically aligned $\mathrm{PVDF}/BaTiO_{3}$ nanopillars	0.0264 ${\mathrm{kPa}}^{-1}$	–	50–600 kPa	–	12,000	[58]
Vertically aligned P(VDF-TrFE) nanopyramids	0.005 $N^{-1}$ (sensor size not specified)	–	–	–	5000	[110]
Single layer interlocked rGO/PVDF nanohemispheres	35 ${\mathrm{kPa}}^{-1}$ below 2.45 kPa	0.6 Pa	0.6 Pa–49.5 kPa	–	5000	[108]

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
