# Peer review of "Innovation Strategy Selection Facilitates High-Performance Flexible Piezoelectric Sensors"

_sensors, 2020, doi:10.3390/s20102820_

Round 1
Reviewer 1 Report
The manuscript reviewed several strategies to improve the performance of piezoelectric sensors. The reviewer thinks it can be accepted to publish.
There should be a space between two words of “which” and “limit” in Page 6, and of "kPa" and "to" in Page 8.
Author Response
Thanks for your comments. We have carefully read the comments, which are all very helpful for revising and improving our review. Moreover, based on the valuable suggestions raised by you, we could try our best to improve the manuscript.
Response to Reviewer 1 Comments
Point 1: There should be a space between two words of “which” and “limit” in Page 6, and of "kPa" and "to" in Page 8.
Response 1: Thank you for your kind correcting. We have carefully checked out our manuscript again and correct all the errors like this.
Reviewer 2 Report
Flexible piezoelectric sensors are indeed attracting high interest nowadays, therefore it is good choice for the topic of a review paper. The manuscript reviews several interesting works in the field, however, the interpretations are often incorrect. Probably the most critical point that the piezoresistive is consequently confused with piezoelectricity in the manuscript.
General comments:
1) Paragraph 2.3 has several incorrect statements, oversimplifications and suggests several misinterpretations on piezoelectricity. Some examples on criticism:
- The role of the atomic size on the piezoelectricity is oversimplified in the manuscript and also shows some contradiction.
- ‘For metallic element dopants, the disordered electric dipoles will be aligned along two electrodes under the external electric field. Consequently, the shielding effect generated by free electrons would be decreased and the piezoelectric properties of ZnO nanowires would be improved’ It is not fully clear.
- ‘As shown in Figure 7a, a strain sensor based on one-dimensional Pt/ZnO hybrid
- nanostructures was developed using a low-temperature, solution-phase method.’ Corresponding reference is missing. But, if it is visible in an SEM image it obviously not a n- or p-type electrical dopant which are atoms distributed in the lattice. The image suggests rather a nanocomposite.
- ‘Besides, conductive nano-materials such as carbon nanotubes[140-143] and graphene[144, 145] are doped into piezoelectric materials to improve the piezoelectric performance.’ Conductive materials do not improve the piezoelectricity! All of the citated references are about piezoresistivity which is a completely different physical property! Good piezoelectric materials are insulators. Or wide band-gap semiconductors. But, any electrical conduction is detrimental.
- ‘For example, as shown in Figure 2d, a flexible films strain sensor was fabricated by dispersing ZnO nanorods and carbon black into PDMS matrix[109]’ It is also pieoresistivity and not piezoelectricity.
- ‘fillers, which can reduce electron loss during the charge flow by forming high conductive channels.’ Why is it beneficial for piezoelectricity?
- ‘ Meanwhile, the addition of conductive nano-materials such as CNT and graphene can increase the conductive channels and make nanoparticles disperse more homogeneously in matrix. Both two methods can improve performance of piezoelectric materials.’ It is not clear either. Why is it beneficial if a piezoelectric material is conductive?
2) The compared sensitivity values are not harmonized in units which makes the comparison troublesome. Even if the cited authors presented these values in different units it would be beneficial to recalculate the results from the published details. Comparative tables are inevitable items of a high quality review paper.
Further comments:
Page 1: which provides new pathways…
Page 2: epsilon0=…… 10−12 F⋅m−1 (unit and exponent are missing)
Page 3: Explanation to LDV: Laser Doppler Vibrometer
Page 3: 72.2 mVkN-1 (space is missing)
Page 4: ‘Up to now ZnO, PZT, BaTiO? and PVDF are the major functional piezoelectric materials….’: The reviewer would also mention AlN as probably one of the the most widely used piezoelectric materials after PZT
Page 6: wearable sensor (with space)
Page 6: ‘its derived copolymers have fabulous properties such as high flexibility because of large length to diameter ratio…’ Does the molecule have large length to diameter ratio? Or the microstructure obtained from PVDF? It is not fully clear.
Page 6: which limits (with space)
Page 6 (and in the whole text): Beta-phase and not ‘pahse’
Page 6: However (with space)
Page 8: ‘1.7 ? ∙ ?-1 to 16.2? ∙ ?-1’, unit of acceleration is m∙ ?-2
Page 8: kPa to (with space)
Page 8: the whole phrase of TMFM and TMCM should be displayed for the non-expert readers
Page 9: ‘which has a large piezoelectric coefficient(?33~112??/?) (as shown in Figure 5d)’ Piezoelectric coefficient is not shown only a stick-and-ball model.
Page 9: ‘Despite the high piezoelectricity and flexibility, strict design and screening strategies, non-universal manufacturing conditions and relatively poor piezoelectricity of uniaxial molecular ferroelectric thin film limit the development of molecular ferroelectrics[100].’ It is difficult to follow this sentence, moreover it seems be a contradiction. Is the piezoelectricity high or low? Or, they refer to different materials?
Page 10: Full phrase for PANI and rGO should be mentioned
Page 10: ‘However, with increasing number of stacked layers, the thickness of the sensor and the consumer of electrons will increase, which, in turn, will inhibit a further development of sensors.’ Consumption of electrons? The meaning of the sentence is not fully clear.
Page 12: ‘mic-morphologies’ micro-morphology? is not instead of isn’t
Page 12: ‘added into piezoelectric materials to increase the piezoelectric coefficient(?33) and the conductive path as well as reduce electron loss’ It is not clear what kind of electron loss is obtained by the dopants? And why is it beneficial for piezoelectric material?
Page 13: 0.38 μW when (with space)
Author Response
Thanks for your comments. We have carefully read the comments, which are all very helpful for
revising and improving our review. Moreover, based on the valuable suggestions raised by you, we could
try our best to improve the manuscript.
As for our responses to your comments, please see the attachment.

Reviewer 3 Report
The paper is well written, however, before the acceptance:
- The authors should consider inserting the traditional equations regards to the direct and reverse effect of the piezo elements;
- The authors need to relate and discuss the challenges between piezoelectric developments and its appliances in SHM, machine failures, industrial process, transformer diagnosis. What these fields are needing about it? The authors mentioned but it needs to be more depth;
- The use of MFCs in transformers and machines need to be inserted in the article. It will improve the work. Look at these articles:
https://ieeexplore.ieee.org/document/8000573
https://ieeexplore.ieee.org/document/4658811
Author Response

(The authors gave the same response as above.)

Round 2
Reviewer 2 Report
The quality of the manuscript has been improved by the revision. Nevertheless, it worth addressing two points.
1) The new paragraph, described in Response 1, is full of grammatical errors. Confusion in punctuation marks, capital and small letters are exchanged, sentence without predicate ('Meanwhile, the corresponding peak of Li-doped ZnO film at 34.24 degrees.'), grammatical errors (would will be substituted) etc. The meaning of the sentence 'The tiny peak shift due to the dopant of Li induced small change of crystal lattice.' is not clear. I think rather the opposite, the crystal lattice change should induce the peak shift'. Even a quick proof-reading before the submission could have eliminated these errors.
2) At the end of 3.1 the authors recommend to apply preamplifier to improve sensitivity, which is well known in the field of piezo transducers. However, it worth emphasizing that the straightforward strategy is the charge sensitive preamplifier instead of voltage or current preamp. The force/pressure dependent piezo signal is the integrated charge (I*t) not the voltage which decays quickly and depends on the cable capacitance.
